# Social Vulnerability in Patients with Multimorbidity: A Cross-Sectional Analysis

**DOI:** 10.3390/ijerph16071244

**Published:** 2019-04-08

**Authors:** Tu N. Nguyen, Patrice Ngangue, Tarek Bouhali, Bridget L. Ryan, Moira Stewart, Martin Fortin

**Affiliations:** 1Faculty of Medicine and Health Sciences, Université de Sherbrooke, Sherbrooke, QC G7H 5H6, Canada; nntu81@gmail.com (T.N.N.); patrice.ngangue@usherbrooke.ca (P.N.); tarek.bouhali@usherbrooke.ca (T.B.); 2Centre for Studies in Family Medicine, Department of Family Medicine, and Department of Epidemiology and Biostatistics, Schulich School of Medicine & Dentistry, Western University, London, ON N6A 3K7, Canada; bryan@uwo.ca (B.L.R.); moira@uwo.ca (M.S.)

**Keywords:** social vulnerability, multimorbidity, primary care

## Abstract

Background: Social aspects play an important role in individual health and should be taken into consideration in the long-term care for people with multimorbidity. Purposes: To describe social vulnerability, to examine its correlation with the number of chronic conditions, and to investigate which chronic conditions were significantly associated with the most socially vulnerable state in patients with multimorbidity. Methods: Cross-sectional analysis from the baseline data of the Patient-Centred Innovations for Persons with Multimorbidity (PACEinMM) Study. Participants were patients attending primary healthcare settings in Quebec, Canada. A social vulnerability index was applied to identify social vulnerability level. The index value ranges from 0 to 1 (1 as the most vulnerable). Spearman’s rank correlation coefficient was calculated for the correlation between the social vulnerability index and the number of chronic conditions. Logistic regression was applied to investigate which chronic conditions were independently associated with the most socially vulnerable state. Results: There were 301 participants, mean age 61.0 ± 10.5, 53.2% female. The mean number of chronic health conditions was 5.01 ± 1.82, with the most common being hyperlipidemia (78.1%), hypertension (69.4%), and obesity (54.2%). The social vulnerability index had a median value of 0.13 (range 0.00–0.78). There was a positive correlation between the social vulnerability index and the number of chronic conditions (r = 0.24, *p* < 0.001). Obesity, depression/anxiety, and cardiovascular diseases were significantly associated with the most socially vulnerable patients with multimorbidity. Conclusions: There was a significant correlation between social vulnerability and the total number of chronic conditions, with depression/anxiety, obesity, and cardiovascular diseases being the most related to social vulnerability.

## 1. Introduction

Social vulnerability has been an emerging topic in recent years. Social vulnerability is a term to provide a holistic and comprehensive approach to measure social circumstances of individuals [1]. It is distinct from socioeconomic status (which is usually operationalized based on education, social status or income) and social determinants of health (which are the conditions in which people are born, grow, live, work, and age) [2,3]. Social vulnerability measured by a social vulnerability index takes into account many aspects of social circumstances [4,5]. There have been several studies reporting the impact of social vulnerability on adverse health outcomes, such as cognitive impairment, disability, and mortality [1,5,6,7]. In the Canadian Study of Health and Aging, increasing social vulnerability defined by using a social vulnerability index was associated with increased risk of cognitive decline during a 5-year follow-up [6]. Another secondary analysis from the Canadian Study of Health and Aging showed that among participants aged 70 or older (*n* = 5,703), those who were more socially vulnerable had a higher 5-year mortality rate compared to those who were the least socially vulnerable (adjusted hazard ratio 2.5, 95% CI 1.5–4.3) [5]. The Honolulu–Asia Aging Study also showed that social vulnerability was associated with increased mortality [7]. In a survey of health, aging, and retirement in ten European countries (the SHARE study), social vulnerability was a significant predictor of mortality (adjusted hazard ratio 1.25, 95% CI 1.07–1.45) and disability (adjusted odds ratio 1.36, 95 % CI 1.15–1.62) after controlling for age, sex, baseline disability, and frailty level [1]. A study of 1,751,841 participants in Scotland also found that the onset of multimorbidity happened 10–15 years earlier in those who were among the most socioeconomically deprived group [8].

According to a recent systematic review on the patterns of multimorbidity in high-income countries, the overall prevalence of multimorbidity was 66.1% [9]. In the United States, data from the National Health and Nutrition Examination Survey (NHANES) showed that the prevalence of multimorbidity was 59.6% (defined as having ≥2 morbidities) and 38.5% (defined as having ≥3 morbidities) [10]. In Canada, the estimated prevalence of multimorbidity ranged from 16.9% to 59.4% in the general population and 29.5% to 69.5% in primary care settings [11]. In recent years, there have been major primary care reforms in Canada and around the world, where multimorbidity is a driver of change, and a holistic view of patients and a generalist approach to care have been recommended [12]. According to the World Health Organization, human health is defined as “a state of complete physical, mental and social well-being and not merely the absence of disease or infirmity”. Social circumstances play an important role in individual health and should be taken into consideration in the long-term care for people with multimorbidity.

In this study, we aim to describe social vulnerability in patients with multimorbidity, to examine its correlation with the number of chronic conditions, and to investigate which chronic conditions were significantly associated with the most socially vulnerable state in patients with multimorbidity.

## 2. Methods

A secondary, cross-sectional analysis was conducted from the baseline data of the Patient-Centred Innovations for Persons with Multimorbidity Study (PACE in MM study) [13]. The PACE in MM study is a mixed-methods study in two Canadian jurisdictions (Quebec and Ontario) that evaluated complex interventions to ameliorate patient-centred outcomes for patients with multimorbidity [13]. This current study used the data from participants recruited in Quebec. Participants were adult patients with multimorbidity attending primary health care settings in Quebec, Canada. Exclusion criteria were those patients with severe cognitive impairment and illiteracy. At baseline, all participants were asked to provide demographic information such as age, gender, height, and weight and asked to respond to questions about chronic health conditions, health status based on the 12-item short-form health survey (SF-12) [14], quality of life based on the EuroQol-5D (EQ-5D) questionnaire [15], psychological well-being with a K6 questionnaire [16], health behaviours with the Health Education Impact Questionnaire (heiQ) [17], and two questionnaires about patient-centred care.

**Chronic health conditions**: For the PACE in MM study, multimorbidity was defined as having ≥3 chronic conditions from a list of 19 self-reported chronic conditions or categories of chronic conditions including hypertension, hyperlipidemia, obesity, diabetes, chronic musculoskeletal conditions causing pain or limitation, arthritis and/or rheumatoid arthritis, osteoporosis, stomach problems (reflux, heartburn, or gastric ulcer), chronic lung disease (asthma, chronic obstructive pulmonary disease, chronic bronchitis), depression/anxiety, cardiovascular disease (angina, myocardial infarction, atrial fibrillation, poor circulation in the lower limbs), heart failure (including valve problems or replacement), colon problem (irritable bowel, Crohn’s disease, ulcerative colitis, diverticulosis), thyroid disorder, any cancer in the previous 5 years (including melanoma but excluding other skin cancer), kidney disease or failure, stroke/transient ischemic attack, chronic urinary problem, and chronic hepatitis [18]. These 19 groups of conditions were used in this current study to obtain a count of chronic health conditions for each participant.

**Social vulnerability definition**: A social vulnerability index proposed by Andrew et al. [4] in Canadian population studies was applied with some adaptations. This index considers six components of vulnerability: communication to engage in the wider community, living situation, social support, social engagement and leisure, empowerment and life control, and socioeconomic status. From the baseline data of PACE in MM, 19 self-reported variables relating to social factors were identified and assigned a score out of a possible 0 to 1 (as in Table 2). Therefore, the total deficit scores range from 0 to 19. These 19 variables showed a good internal consistency (Cronbach’s alpha = 0.75). A social vulnerability index was calculated by dividing the total scores by 19, resulting in an index value ranging from 0 to 1, with 1 as the most vulnerable. The index was then further categorized into quintiles.

### 2.1. Ethics Approval

The PACE in MM study was approved by the Research Ethics Board of the Integrated University Health and Social Services Centre of Saguenay-Lac-St-Jean (Comité d’éthique de la recherche du Centre intégré universitaire de santé et de services sociaux du Saguenay–Lac-St-Jean) (Ethical code 2013-010).

### 2.2. Statistical Analysis

Analysis of the data was performed using SPSS for Windows 24.0 (IBM Corp., Armonk, NY, USA). Continuous variables are presented as mean ± standard deviation or median (range) and categorical variables as frequencies and percentages. Two-tailed *p*-values < 0.05 were considered statistically significant.

To examine the correlation between social vulnerability and the number of chronic conditions, ANOVA was conducted to determine if the mean number of chronic conditions varied by social vulnerability index. Spearman’s rank correlation coefficient was also calculated for the association between the social vulnerability index and the number of chronic conditions. Two-tailed *p*-values < 0.05 were considered statistically significant.

Logistic regression was applied to investigate which type of chronic condition was significantly associated with the most socially vulnerable state. For this analysis, the outcome compared those participants in quintile 5 (the most socially vulnerable state) to those in quintiles 1 to 4 (representing the less socially vulnerable states). Univariate logistic regression was performed on age, gender, and each chronic condition in the list. Variables that had a *p*-value < 0.20 on univariate analysis were selected for multivariate analysis. A backward elimination method was applied, and the final model only retained variables significant at *p* < 0.05. Results are presented as odds ratios (OR) and 95% confidence intervals (CIs).

## 3. Results

A total of 301 participants were eligible for this study, mean age 61.0 ± 10.5, 53.2% female. The most common chronic health conditions were hyperlipidemia (78.1%), hypertension (69.4%), and obesity (54.2%) (Table 1).

### 3.1. Description of the Social Vulnerability Index in Participants with Multimorbidity

The individual items that were used to construct the social vulnerability index are presented in Table 2. The social vulnerability index had a mean value of 0.16, median value of 0.13 (range 0.00–0.78), and quintiles as follows: (1st quintile-lowest vulnerability) social vulnerability index ≤0.052, (2nd quintile) social vulnerability index from 0.053 to 0.109, (3rd quintile) social vulnerability index from 0.110 to 0.159, (4th quintile) social vulnerability index from 0.160 to 0.259, and (5th quintile-highest vulnerability) social vulnerability index ≥0.260. Among 301 participants with multimorbidity, 36 (12.0%) were free of social deficits (their social vulnerability index was zero).

### 3.2. The Relationship between Social Vulnerability and Number of Chronic Health Conditions

There were statistically significant differences in the mean number of chronic health conditions among the quintiles of the social vulnerability index: Quintile 1 (lowest vulnerability)—4.4 ± 1.7; Quintile 2—4.9 ± 1.8; Quintile 3—5.1 ± 1.8; Quintile 4—5.2 ± 1.8; and Quintile 5 (highest vulnerability)—5.6 ± 2.0 (as determined by one-way ANOVA, F (4270) = 3.3, *p* = 0.01).

There was a significantly positive correlation between the social vulnerability index and the number of chronic conditions (r = 0.24, *p* < 0.001).

### 3.3. The Relationship between Chronic Condition Types with the Most Socially Vulnerable State

From univariate logistic regression analysis, variables with a *p*-value < 0.20 were selected for multivariate logistic regression (obesity, cardiovascular disease, heart failure, chronic lung disease, depression/anxiety, osteoporosis, any cancer in the previous 5 years). A backward elimination method was applied, and the final model retained depression/anxiety (adjusted OR 2.28, 95% CI 1.25–4.15), obesity (adjusted OR 2.74, 95% CI 1.43–5.27), and cardiovascular diseases (adjusted OR 2.38, 95% CI 1.17–4.84) (Table 3).

## 4. Discussion

In this study of 301 participants with multimorbidity, we applied a social vulnerability index to investigate the relationship between social vulnerability and chronic health conditions. The increased social vulnerability level was associated with increased number of chronic conditions, and among the list of 19 chronic conditions, obesity, depression/anxiety, and cardiovascular diseases were independently associated with the most socially vulnerable state in patients with multimorbidity.

According to the previous study of Andrew et al. in older Canadians [4], the median social vulnerability index was 0.25 in the Canadian Study of Health and Aging and 0.28 in the National Population Health Survey. Compared to these values, the median social vulnerability index in our study was lower. It may be due to the difference in age (our study included participants aged from 26 to 80, with 62.5% participants aged less than 65 years old), and all participants in our study were patients with multimorbidity recruited for an intervention study in primary care.

Although there has been no published study on the relationship between multimorbidity and social vulnerability defined by a social vulnerability index, several studies have reported the negative impact of socioeconomic deprivation on multimorbidity. People with lower socioeconomic condition were more likely to develop multimorbidity, and the onset of multimorbidity was also likely to be significantly earlier [8,19,20]. In a study from the Canadian Community Health Survey on 105,416 Canadian adults, those with multimorbidity (having 3 or more chronic conditions) were more likely to be living in the lowest income quintile and to have not completed high school [19]. In another study in 1,751,841 patients in Scotland, people with the most socioeconomic deprivation had an earlier onset of multimorbidity (around 10 to 15 years) compared to the least socioeconomic deprivation [8].

Among the list of 19 chronic conditions, we found that depression/anxiety, obesity, and cardiovascular diseases were significantly associated with the most socially vulnerable state in patients with multimorbidity (having at least three chronic conditions). As social engagement and empowerment/life control are two components of the social vulnerability index, it can be expected that patients with depression/anxiety will be at higher risk of being socially vulnerable. There has been consistent evidence about the relationship between low socioeconomic condition and elevated cardiovascular risk [21,22,23,24]. People with socioeconomic disadvantage are prone to smoking, heavy drinking, obesity, physical inactivity, and may have limited accessibility to healthy foods and preventive care [23], which can contribute to the development of cardiovascular diseases. Many studies also reported the increased prevalence of multimorbidity in people with obesity [20,25,26,27], and obesity is on the rise in socioeconomically disadvantaged populations [28].

The evidence from this study may support the use of a social vulnerability index in evaluating social aspects of patients with multimorbidity in primary care. Compared to the model suggested by Andrew et al. (4), where the social vulnerability index was constructed based on 40 items for the Canadian Study of Health and Aging and 23 items for the National Population Health Study, in our study, the social vulnerability index was constructed based on 19 items and showed a good internal consistency. The major limitation of this study is that it was based on self-reported information of chronic health conditions; hence, there may be some bias in recall of the participants’ medical history. In addition, this study was a secondary analysis, and so, we were limited by the present questionnaires in the social vulnerability items we identified. Further, the study may be limited by sample size, particularly the small number of people with some less common chronic conditions, such as hepatitis.

## 5. Conclusions

In this study, there was a significant correlation between social vulnerability and total number of chronic conditions. In patients with multimorbidity, those with depression/anxiety, obesity, and cardiovascular diseases were the most socially vulnerable. As the number of people living with multimorbidity is increasing, recognizing patients with social vulnerability is crucial for patient–centred care. In addition to standard medical care, these patients need more aids in other social aspects to ensure better treatment outcomes.

## Figures and Tables

**Table 1 ijerph-16-01244-t001:** General characteristics.

Variables	N = 301
Age (years)	61.0 ± 10.5
Female	160 (53.2)
BMI (kg/m^2^)	31.55 ± 6.41
Number of chronic health conditions	5.01 ± 1.82
*Prevalence of chronic health conditions:*	
Hyperlipidemia	235 (78.1)
Hypertension	209 (69.4)
Obese (BMI ≥ 30)	162 (54.2)
Chronic musculoskeletal conditions causing pain or limitation	154 (51.2)
Diabetes	149 (49.5)
Stomach problem (reflux, heartburn, or gastric ulcer)	111 (36.9)
Asthma, COPD, or chronic bronchitis	96 (31.9)
Depression/anxiety	91 (30.2)
Cardiovascular disease (angina, myocardial infarction, atrial fibrillation, poor circulation	64 (21.3)
in the lower limbs)	
Colon problem (irritable bowel, Crohn’s disease, ulcerative colitis, diverticulosis)	43 (14.3)
Thyroid disorder	37 (12.3)
Arthritis and/or rheumatoid arthritis	26 (8.6)
Osteoporosis	26 (8.6)
Any cancer in the previous 5 years (including melanoma, but excluding other skin cancer)	22 (7.3)
Kidney disease or failure	20 (6.6)
Stroke/Transient ischemic attack	17 (5.6)
Chronic urinary problem	14 (4.7)
Heart failure (including valve problems or replacement)	11 (3.7)
Chronic hepatitis	1 (0.3)

Continuous data are presented as mean ± standard deviation. Categorical data are shown as *n* (%).

**Table 2 ijerph-16-01244-t002:** Items of the social vulnerability index.

	N = 301
*Communication to engage in wider community:*	
**1. Can speak English or French:**	
No (1 point)	0 (0)
Yes (0 point)	301 (100)
*Living situation:*	
**2. Marital status:**	
Single/divorced/widower (1 point)	104 (34.6)
Married (0 point)	197 (65.4)
*Social support:*	
**3. If I need help, I have plenty of people I can rely on**	
Answers of “strongly disagree” or “disagree” (1 point)	21 (7.0)
Answers of “strongly agree” or “agree” (0 point)	280 (93.0)
**4. When I feel ill, my family and carers really understand what I am going through**	
Answers of “strongly disagree” or “disagree” (1 point)	40 (13.3)
Answers of “strongly agree” or “agree” (0 point)	260 (86.7)
**5. I have enough friends who help me cope with my health problems**	
Answers of “strongly disagree” or “disagree” (1 point)	53 (17.7)
Answers of “strongly agree” or “agree” (0 point)	247 (82.3)
**6. Overall, I feel well looked after by friends or family**	
Answers of “strongly disagree” or “disagree” (1 point)	16 (5.3)
Answers of “strongly agree” or “agree” (0 point)	285 (94.7)
*Social engagement and leisure:*	
**7. During the past four weeks, how much of the time has your physical health or emotional problems interfered with your social activities (like visiting with friends, relatives, etc.)?**	
all of the time (1 point)	5 (1.7)
most of the time (0.75 point)	21 (7.0)
some of the time (0.50 point)	57 (18.9)
a little of the time (0.25 point)	65 (21.6)
none of the time (0 point)	153 (50.8)
**8. I feel like I am actively involved in life**	
Answers of “strongly disagree” or “disagree” (1 point)	22 (7.3)
Answers of “strongly agree” or “agree” (0 point)	278 (92.7)
**9. On most days of the week, I do at least one activity to improve my health**	
Answers of “strongly disagree” or “disagree” (1 point)	66 (21.9)
Answers of “strongly agree” or “agree” (0 point)	235 (78.1)
**10. I do at least one type of physical activity every day for at least 30 minutes**	
Answers of “strongly disagree” or “disagree” (1 point)	106 (35.2)
Answers of “strongly agree” or “agree” (0 point)	195 (64.8)
**11. I have plans to do enjoyable things for myself during the next few days**	
Answers of “strongly disagree” or “disagree” (1 point)	4 (1.3)
Answers of “strongly agree” or “agree” (0 point)	297 (98.7)
**12. On most days of the week, I set aside time for healthy activities**	
Answers of “strongly disagree” or “disagree” (1 point)	95 (31.6)
Answers of “strongly agree” or “agree” (0 point)	206 (68.4)
**13. I walk for exercise, for at least 15 minutes per day, most days of the week**	
Answers of “strongly disagree” or “disagree” (1 point)	140 (46.5)
Answers of “strongly agree” or “agree” (0 point)	161 (53.5)
*Empowerment, life control:*	
**14. I try to make the most of my life**	
Answers of “strongly disagree” or “disagree” (1 point)	5 (1.7)
Answers of “strongly agree” or “agree” (0 point)	296 (98.3)
**15. I am doing interesting things in my life**	
Answers of “strongly disagree” or “disagree” (1 point)	13 (4.3)
Answers of “strongly agree” or “agree” (0 point)	288 (95.7)
**16. I feel I have a very good life even when I have health problem**	
Answers of “strongly disagree” or “disagree” (1 point)	16 (5.3)
Answers of “strongly agree” or “agree” (0 point)	285 (94.7)
**17. I do not let my health problems control my life**	
Answers of “strongly disagree” or “disagree” (1 point)	23 (7.6)
Answers of “strongly agree” or “agree” (0 point)	278 (92.4)
*Socioeconomic status:*	
**18. Income level**	
Low income (<20000) (1 point)	56 (19.4)
Income ≥20000 (0 point)	232 (80.6)
**19. Education level**	
Did not complete secondary school (1 point)	68 (22.6)
Complete secondary school or higher (0 point)	233 (77.4)

Data are shown as *n* (%).

**Table 3 ijerph-16-01244-t003:** Factors associated with the most social vulnerability on univariate and multivariate logistic regression (*n* = 301).

Variables	Univariate		Multivariable	
	Odds Ratio for Being the Most Socially Vulnerable (95% CI)	*p*	Adjusted Odds Ratio for Being the Most Socially Vulnerable (95% CI)	*p*
Age	0.99 (0.97–1.02)	0.735	–	–
Female gender	1.35 (0.76–2.38)	0.305	–	–
Hypertension	1.45 (0.76–2.76)	0.258	–	–
Hyperlipidemia	1.05 (0.53–2.08)	0.896	–	–
Obesity	2.31 (1.26–4.25)	0.007	2.74 (1.43–5.27)	0.002
Diabetes	1.16 (0.66–2.04)	0.605	–	–
Cardiovascular disease (angina, myocardial infarction, atrial fibrillation, poor circulation in the lower limbs)	1.59 (0.83–3.02)	0.160	2.38 (1.17–4.84)	0.017
Heart failure (including valve problems or replacement)	2.34 (0.66–8.25)	0.188	–	–
Stroke/transient ischemic attack	1.23 (0.39–3.90)	0.731	–	–
Chronic lung diseases	1.65 (0.93–2.95)	0.090	–	–
Depression/anxiety	2.38 (1.34–4.26)	0.003	2.28 (1.25–4.15)	0.007
Thyroid disorder	1.31 (0.58–2.95)	0.513	–	–
Chronic musculoskeletal conditions causing pain or limitation	1.07 (0.61–1.87)	0.821	–	–
Arthritis and/or rheumatoid arthritis	1.20 (0.46–3.13)	0.709	–	–
Osteoporosis	2.27 (0.96–5.38)	0.062	–	–
Any cancer in the previous 5 years (including melanoma, but excluding other skin cancer)	0.37 (0.08–1.64)	0.192	–	–
Kidney disease or failure	0.98 (0.32–3.05)	0.976	–	–
Chronic urinary problem	1.08 (0.29–3.99)	0.912	–	–
Stomach problem (reflux, heartburn, or gastric ulcer)	0.96 (0.53–1.72)	0.883	–	–
Colon problem (irritable bowel, Crohn’s disease, ulcerative colitis, diverticulosis)	0.88 (0.39–2.02)	0.770	–	–
Chronic hepatitis	N/A due to small number (n = 1)		–	–

Only variables that had a *p*–value < 0.20 in univariate regression were entered into multiple regression model. The final model contains only variables with *p*-value < 0.05.

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
