# Peer review of "Social Vulnerability in Patients with Multimorbidity: A Cross-Sectional Analysis"

_ijerph, 2019, doi:10.3390/ijerph16071244_

Round 1

Reviewer 1 Report

Very interesting work. Here are a few recommandations: The introduction should define the concept of vulnerability and further present its importance in relation to the reported problem. Social vulnerability should be distinguished from socio-economic disadvantage and social determinants of health. 

Author Response

Very interesting work. Here are a few recommendations: The introduction should define the concept of vulnerability and further present its importance in relation to the reported problem. Social vulnerability should be distinguished from socio- economic disadvantage and social determinants of health.

Response:

Thank you for your suggestion. We have added more information in the introduction part (lines 33-39)

Reviewer 2 Report

This is a very well written paper based on the data of cohort study.

Some comments:

Chronic health condition seems to be more adequate term rather than chronic disease. For ex.  Chronic musculoskeletal conditions causing pain or limitation is a condition not specific disease

If then, the terminology of multimorbidity may be changed, too.

   2. Authors mentioned in conclusion role of social vulnerability in the development of

 multimorbidity, especially in those with depression/anxiety, obesity and cardiovascular diseases

As this is a cross-sectional study, we can not say which one is cause, which one is result.

So I would recommend amendment of the sentence and also address the limitation of cross-sectional study

   3. By chance, Cancer was related with lower risk of social vulnerability index. How do you explain that?

Author Response

This is a very well written paper based on the data of cohort study. Some comments:

Chronic health condition seems to be more adequate term rather than chronic disease. For ex. Chronic musculoskeletal conditions causing pain or limitation is a condition not specific disease.

If then, the terminology of multimorbidity may be changed, too.

Response: Thank you for your comments. Both terms “chronic disease” and “chronic condition” have been used interchangeably in previous publications and in this manuscript.

https://www.ncbi.nlm.nih.gov/pubmed/28947426

https://www.ncbi.nlm.nih.gov/pubmed/28487349

However, for the sake of clarity, we changed the term disease for condition that is more inclusive except for the specific diseases mentioned in the list of conditions or group of chronic conditions where the term disease is more applicable (Example: cardiovascular diseases, Crohn disease, etc.)

Authors mentioned in conclusion “role of social vulnerability in the development of multimorbidity, especially in those with depression/anxiety, obesity and cardiovascular diseases”

As this is a cross-sectional study, we cannot say which one is cause, which one is result.

So I would recommend amendment of the sentence and also address the limitation of cross-sectional study

Response: We have revised the conclusion (lines 205-209).

By chance, Cancer was related with lower risk of social vulnerability index. How do you explain that?

Response: This result came from a univariate logistic regression analysis and the p value did not reach statistical significance.

Reviewer 3 Report

Taking into account the results obtained, the conclusion is general and scarce. It is recommended to include some more precise conclusion based on the results.

Author Response

Taking into account the results obtained, the conclusion is general and scarce. It is recommended to include some more precise conclusion based on the results.

Response: Thank you for your suggestion. We have revised the conclusion (lines 205-209).